# Histone Deacetylases in Retinoblastoma

**DOI:** 10.3390/ijms25136910

**Published:** 2024-06-24

**Authors:** Malwina Lisek, Julia Tomczak, Julia Swiatek, Aleksandra Kaluza, Tomasz Boczek

**Affiliations:** Department of Molecular Neurochemistry, Medical University of Lodz, 90-419 Lodz, Poland; julia.tomczak1@student.umed.lodz.pl (J.T.); julia.swiatek1@student.umed.lodz.pl (J.S.); aleksandra.kaluza@student.umed.lodz.pl (A.K.)

**Keywords:** retinoblastoma, histone deacetylase inhibitors, chromatic remodeling, gene expression, cancer treatment therapies

## Abstract

Retinoblastoma, a pediatric ocular malignancy, presents significant challenges in comprehending its molecular underpinnings and targeted therapeutic approaches. The dysregulated activity of histone deacetylases (HDACs) has been associated with retinoblastoma pathogenesis, influencing critical cellular processes like cell cycle regulation or retinal ganglion cell apoptosis. Through their deacetylase activity, HDACs exert control over key tumor suppressors and oncogenes, influencing the delicate equilibrium between proliferation and cell death. Furthermore, the interplay between HDACs and the retinoblastoma protein pathway, a pivotal aspect of retinoblastoma etiology, reveals a complex network of interactions influencing the tumor microenvironment. The examination of HDAC inhibitors, encompassing both established and novel compounds, offers insights into potential approaches to restore acetylation balance and impede retinoblastoma progression. Moreover, the identification of specific HDAC isoforms exhibiting varying expression in retinoblastoma provides avenues for personalized therapeutic strategies, allowing for interventions tailored to individual patient profiles. This review focuses on the intricate interrelationship between HDACs and retinoblastoma, shedding light on epigenetic mechanisms that control tumor development and progression. The exploration of HDAC-targeted therapies underscores the potential for innovative treatment modalities in the pursuit of more efficacious and personalized management strategies for this disease.

## 1. Introduction to Retinoblastoma

Retinoblastoma (RB), the most prevalent malignant intraocular tumor in children, predominantly affects patients before the age of three, originating from the cones of the retina, which possess certain traits making them susceptible to tumorigenesis. Previous research documented notable variations in retinoblastoma incidence, linked to factors like gender, ethnicity, and exposure to unsanitary conditions. However, recent studies question the significance of these disparities, suggesting a consistent global occurrence of retinoblastoma. The estimated prevalence of this disease in the global population is approximately 1 in 16,000 to 20,000 live births. In the United States alone, there are roughly 200–300 new cases reported annually (approximately 8000 worldwide), a trend that has persisted for over four decades [1,2,3]. According to a comprehensive study by Dimaras and colleagues [4], 11% of affected children reside in high-income countries, 69% in middle-income countries, and 20% in low-income countries. Despite its prevalence in low- and middle-income nations, the majority of research and treatment facilities are situated in middle- to high-income regions, resulting in disparities in healthcare access. Due to significant economic differences, retinoblastoma is associated with lower patient survival rates (~30%) in low-income countries compared to the almost complete curability in high-income countries (>95%).

Nearly all retinoblastomas develop following biallelic inactivation of RB1 gene, located in humans on chromosome 13—more specifically, 13q14.1-q14.2. Human RB1 gene is translated into 928 amino acid protein (pRB), which is a tumor suppressor protein belonging to the pocket protein family. The pRB is primarily recognized as a controller of the cell cycle through interaction with E2F transcription factors to suppress genes associated with cell proliferation. When stimulated by mitogenic signals, cyclin-dependent kinases (CDKs) hyperphosphorylate pRB, releasing its repression and facilitating the transition from G1 to S phase (Figure 1). The absence of pRB, but also several modifications impairing its function, for instance mutations, deletions, or promoter methylation, may result in the alleviation of this repression even in the absence of mitogenic signals, allowing cells to enter the cell cycle [5,6]. Based on the canonical model of negative regulation of E2F transcription factor by pRB, it is tempting to speculate that loss of pRB function would be a primary reason for retinoblastoma development [7,8]. However, E2F transcription can also be repressed by p107 and p130, which are pRB-related proteins involved in cell cycle regulation, indicating that some cancer types may restrict E2F activation [9]. pRB also enhances the expression of p27, an inhibitor of cyclin-dependent protein kinases, which is also involved in the inhibition of cell cycle progression. However, as cone precursor maturation progresses, elevated levels of pRB coincide with reduced p27 expression, suggesting that p27 may not play a role in suppressing retinoblastoma in these cells [10]. Interestingly, the study of Chen and colleagues demonstrated that E2F1-3 null retinal progenitor cells or activated Müller glia can divide normally, suggesting that E2Fs may be dispensable for in vivo proliferation, at least for certain types of cells [11]. In view of this finding, it is difficult to understand how the pRB-mediated inhibition of these E2Fs can inhibit proliferation if they are not required for proliferation. These findings imply that transcriptional regulation of E2Fs might not be the sole mechanism through which RB acts as a tumor suppressor.

Additionally, pRB interacts with chromatin remodeling proteins like histone deacetylases (HDACs), leading to modifications in chromatin structure and gene expression that contribute to the development of tumors [12]. Besides the dysregulation of the pRB signaling pathway, retinoblastoma’s oncogenic properties are also influenced by other pathways like p53, Wnt, and Ras/MEK/ERK. These pathways contribute to retinoblastoma’s tumorigenic effects like development of stem cell-like features, triggering epithelial–mesenchymal transition (EMT) and causing metabolic reprogramming [13]. Therefore, the exact mechanism by which pRB typically suppresses retinoblastoma has yet to be completely understood.

The diagnosis of retinoblastoma is typically confirmed through a fundus examination of the eye using indirect ophthalmoscopy. The most common initial signs are leukocoria, followed by strabismus (misaligned eyes) when central vision is affected. Without timely treatment, the disease may progress to an advanced stage, characterized by changes in iris color, enlargement of the cornea and eye due to increased pressure, or non-infectious inflammation in the orbital region. Advanced retinoblastoma stages can also lead to exophthalmos (protruding eyeball) and swelling of the soft tissues around the eye [4]. Additional imaging techniques such as MRI (magnetic resonance imaging) are often used in clinical practice to assess optic nerve involvement or the presence of primitive neuroectodermal tumors associated with RB1 mutations [14,15,16].

Nearly all retinoblastomas develop after both RB1 alleles are deactivated in a cone cell precursor during retinal development. However, the mere loss of RB1 does not seem to be enough to initiate tumor malignancy, as benign retinomas also display loss of both RB1 alleles [17]. Several reports suggest that the development of retinoblastoma in individuals with RB1 loss may also be driven by additional factors such as epigenetic alterations (including histone H3K4 methylation and H3K9/H3K14 acetylation), copy number variations (in genes such as Kif14, Mdm4, E2f3, Dek, and Cdh11), or inactivating mutations in the BCL-6 co-repressor [18,19]. Analysis of gene expression profiles of 21 retinoblastomas has shown a segregation into two subtypes: tumors with invasive growth expressing genes associated with various retinal cell types, and tumors resembling cone photoreceptor cells, characterized by high metabolic activity [20]. Although mutations in both RB1 alleles appear to be central to retinoblastoma, a small subset of unilateral tumors lack detectable mutations in this gene; instead, they exhibit significantly increased expression of the n-myc proto-oncogene (MYCN—myelocytomatosis-neuroblastoma) [21]. MYCN-amplified retinoblastomas exhibit distinct gene signatures associated with MYCN overexpression, acting as the primary driver in this rare and aggressive subtype [22,23].
Figure 1Cell cycle control by pRb and HDAC by regulation of E2F-dependent gene expression. pRb forms complexes with E2F transcriptional regulators bounded to DNA and inhibit their target gene expression by recruiting HDACs, co-repressors, and enzymes responsible for chromatin remodeling. The regulation of E2F-dependent gene expression by pRb is crucial for controlling cell cycle progression, maintaining genomic stability, and preventing aberrant cell proliferation that could lead to cancer [5,6,7,8]. Figure based on [24].
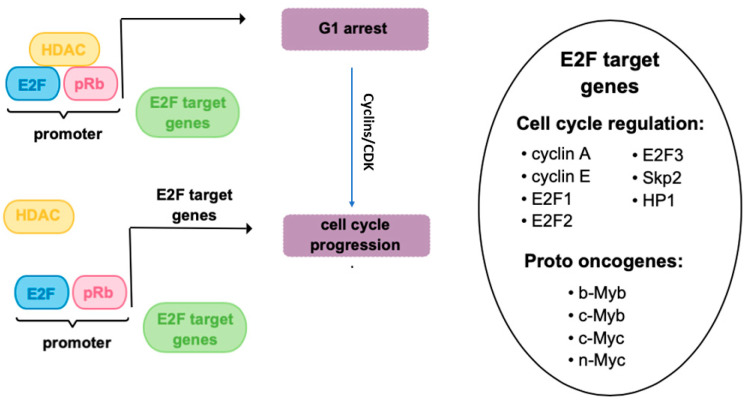



## 2. Retinoblastoma Pre-Clinical Models

The widespread expression of pRB in nearly all cells makes the particular vulnerability of the retina to pRB loss puzzling. While this sensitivity likely stems from the characteristics of the cell from which retinoblastoma originates, pinpointing this cell type has been challenging. Preclinical models are essential for understanding the cell of origin and tumor development mechanisms and establishing potential treatments. In vitro testing using cell lines and primary tumor cells is widely adopted due to its straightforward implementation, cost-effectiveness, and rapid provision of results. The establishment of the initial human retinoblastoma cell line, Y79, represented a significant milestone in retinoblastoma research. This breakthrough was followed by the development of subsequent cell lines such as WERI-Rb1, which have played a pivotal role in further advancing the field [25,26]. Y79 cells exhibit hypertriploidy with variable numbers of chromosomes and rapid growth rates, rendering them suitable for experiments aimed at suppressing tumor growth. WERI-Rb1 cells possess morphological similarities but tend to grow as loose cell aggregates, demonstrating low levels of endogenous cell death [27]. Notably, Y79-derived tumors in xenograft models mimic invasive and metastatic disease, contrasting with the non-metastatic ocular tumors produced by WERI-Rb1 [28]. Moreover, other cell lines derived from unilateral and bilateral retinoblastomas have provided valuable insights into the neuronal phenotypes associated with different RB1 gene mutations and/or additional non-RB1 mutations [29,30]. Resistance to drugs and the recurrence of cancer after treatment are primary concerns associated with chemotherapy, which remains the primary method for preserving the eye in retinoblastoma treatment. Creation of etoposide- and cisplatin-resistant Y79 and WERI-Rb1 cell lines has provided valuable models for studying chemoresistance mechanisms. These resistant cell lines exhibit increased growth kinetics and apoptotic rates, offering insights into potential therapeutic strategies [31]. In a recent study, Jahagirdar et al. have described a triple co-culture system involving retinoblastoma cells, retinal epithelial cells, and choroid endothelial cells. This model, established using a protein coating cocktail, aimed to replicate the complex interactions within the ocular environment under in vitro conditions. By incorporating multiple cell types relevant to retinoblastoma, this model provided a comprehensive platform for studying the disease and evaluating potential therapeutic interventions [32].

In recent years, there has been a significant transition towards utilizing three-dimensional (3D) cell culture models in retinoblastoma research due to limitations of traditional 2D cultures in capturing disease complexity and drug responses accurately. An ideal 3D culture system for retinoblastoma should faithfully replicate the pathophysiological features of the disease, encompassing genomic traits, gene/protein expression profiles, and responses to therapeutic agents [30]. It should also accommodate various treatment approaches and be scalable for large drug screenings [31]. Techniques for generating 3D cell cultures in retinoblastoma research include scaffold-free and scaffold-based methods, such as mono- and multicellular cultures, low attachment plates, hanging drop method, bioprinting, and microfluidic systems, each offering unique advantages for specific research objectives or applications [27].

Modifying the cellular environment is also essential for disease modeling in retinoblastoma. For instance, materials like Matrigel^®^, which resemble basement membranes, have been utilized to promote the growth of retinoblastoma cells and maintain their histological features and drug responses [33]. Scaffold-based models, including those using polymeric microparticles or magnetic levitation, offer opportunities for high-throughput drug screening but may lack certain physiological features present in vivo [34]. Retinal organoids generated from stem cells represent another exciting avenue for retinoblastoma disease modeling and drug screening. These organoids can be differentiated into multiple cell types found in the retina and can replicate disease processes such as retinoblastoma tumorigenesis [30]. Nonetheless, challenges like extended maturation periods and maintenance requirements presently constrain their extensive adoption for routine drug screening. In recent years, the chick embryo chorioallantois membrane (CAM) assay has gained popularity as an in vivo 3D animal model for various cancers [35]. Specifically in retinoblastoma research, the CAM model has become invaluable for investigating tumor growth, metastasis and treatment strategies. With its partial immune deficiency, the CAM allows for analyzing the growth of different tumor types without interference from specific or nonspecific immune responses. Notably, the CAM’s robust blood vessel network supports the survival, proliferation, and swift vascularization of tumors derived from cancer cell inoculations or xenografted tissues [36,37,38,39]. Utilizing recognized cell lines like WERI-Rb1 [40], Y-79, RB 383, and RB355 [27] enables the study of tumor dynamics and metastatic behaviors under controlled conditions by implanting these cells onto the CAM. RB cells derived from primary tumors can be expanded and cultured on the CAM to assess various drugs, identifying tumor sensitivity and potential responses [41]. Despite the promise of 3D cell culture models in retinoblastoma research, challenges such as sample variability, limited access to patient-derived materials, and scalability issues persist, hindering large-scale drug screening efforts. Moreover, optimizing culture conditions and devising standardized protocols are imperative to guarantee the reproducibility and reliability of findings across various laboratories [42].

Retinoblastoma cell lines have enabled researchers to transplant retinoblastoma cells into immunodeficient animal models through microinjection, leading to tumor formation. Early studies involved injecting retinoblastoma specimens obtained after enucleation or cultured Y79 cells into athymic nude mice and albino CDF rats [43,44]. Y79 cells exhibited a metastatic phenotype, invading various structures including the retina, subretinal space, choroid, optic nerve head, and anterior chamber of the eye, and progressing into the subarachnoid space with focal brain invasion. In contrast, WERI cells formed tumors confined to the eye, with anterior choroidal invasion only occurring at advanced stages [25]. These pioneering investigations laid the foundation for further progress in xenograft modeling of retinoblastoma. Notably, xenografts in rabbits exhibited distinct characteristics, including heightened vascularization, sustained tumor growth and the manifestation of necrotic areas and hypoxic conditions, closely mirroring features observed in human retinoblastomas [45]. In retinoblastoma research, two primary types of xenograft models are utilized: subcutaneous and orthotopic models [46]. Subcutaneous xenograft models involve implanting tumor cells or tissue fragments into the subcutaneous tissue of immunodeficient mice. While subcutaneous tumor growth can be achieved in any immunodeficient mouse strain, it is most effective in severe combined immunodeficiency (SCID) mice, which lack both B and T lymphocytes [47].

These models are relatively simple to establish and allow for in vivo growth monitoring using calipers or imaging techniques such as ultrasound, computed tomography, or magnetic resonance imaging [46]. However, they may not accurately replicate the tumor microenvironment and drug penetration characteristics of intraocular tumors [44]. The orthotopic models entail injecting tumor cells directly into the eye, either into the vitreous or subretinal space, to simulate the native ocular environment. These models can be established in both immunodeficient and immunocompetent mice without requiring immunosuppressive drugs, likely because of the eye’s immune-privileged status [48].

Although challenging to establish, orthotopic models provide a more faithful depiction of intraocular tumor growth and drug response. This attribute is especially beneficial in xenograft models, guaranteeing robust tumor growth and simplifying the evaluation of therapeutic interventions [49]. The success of establishing an orthotopic mouse model for retinoblastoma hinges on the choice of engrafted cells (cell lines or patient-derived xenografts) and the xenografted mouse strain. SCID mice show the highest tumor engraftment rates due to their immunodeficiency. Severe combined immunodeficiency mice, being immunodeficient, exhibit the highest tumor engraftment rate. However, intraocular tumor growth is also satisfactory in the immunocompetent strain B6Albino, offering the potential to investigate the immune response following tumor treatment [50]. Notably, xenografts in rabbits exhibited distinct characteristics, including heightened vascularization, sustained tumor growth, and the manifestation of necrotic areas and hypoxic conditions, closely mirroring features observed in human retinoblastomas [42]. The rabbit xenograft model, established by injecting retinoblastoma cells intravitreally into cyclosporine-immunosuppressed rabbits, has been utilized to evaluate the efficacy and toxicity of intravitreal chemotherapies, such as topotecan and melphalan [51]. Furthermore, the rabbit orthotopic xenograft model of retinoblastoma vitreous seeds has been critical in developing protocols for determining vitreous pharmacokinetics, assessing drug cytotoxicity, calculating optimal doses, determining maximum tolerable doses, and evaluating drug efficacy [52].

Although xenograft models provide valuable insights, they have inherent limitations. For instance, they may not entirely replicate the complexity of the human disease and the immunodeficient nature of the animal hosts complicates effective monitoring of the tumor microenvironment [53]. Moreover, differences in biological species, such as variations in optimal body temperature, can complicate model development and the interpretation of results. To overcome these challenges, researchers have devised innovative approaches. One such strategy involves conducting preclinical experiments with zebrafish at physiological temperatures [54]. By utilizing immune-deficient zebrafish strains, researchers can establish tumor xenografts and study tumor biology under conditions that more closely resemble human physiology. This advancement enhances the relevance and accuracy of xenograft models in retinoblastoma research [55]. The zebrafish model is beneficial for retinoblastoma research due to its rapid development, low maintenance costs, and transparency, which allow for long-term microscopy analyses [56]. Mauricic et al. developed a zebrafish retinoblastoma model showing the migration of three retinoblastoma cell lines (RB355, WERI-RB-1, Y79) into the brain and brain ventricles, mimicking human metastasis patterns. Despite challenges in drug testing due to limited fluorescence duration, H&E staining can extend observation periods [57]. Additionally, implanting human retinoblastoma cells into newborn rat eyes provides a mammalian orthotopic model for studying retinoblastoma’s spatiotemporal development and tumor growth using bioluminescence imaging and optical coherence tomography [58]. Combining these models offers complementary insights into retinoblastoma progression and therapy development [59].

The versatility of retinoblastoma models is crucial due to the diverse nature of human retinoblastoma, ranging from solid retinal tumors to diffuse vitreous tumors. In transgenic models, tumors develop from the retina, while orthotopic xenograft models may display both retinal and intravitreal tumor growth. Notably, tumor growth in the vitreous, even with subretinal injections, highlights the need to evaluate therapies targeting vitreous seeds, utilizing intravitreal tumor growth in mouse models for testing. However, challenges in visualization during specific examinations may favor genetically engineered mouse models (GEMMs), which predominantly develop retinal tumors, enabling clearer visualization [46]. GEMMs have become indispensable in retinoblastoma research, shedding light on tumorigenesis, progression, and potential treatments [53]. The development of GEMMs for retinoblastoma began with efforts to recapitulate the genetic alterations observed in human tumors. One of the earliest transgenic models, the LH-beta T-Ag model expressed the oncogenic SV40 early region under the control of the luteinizing hormone β subunit (LHβ) promoter in the gonadotropic cells of the anterior pituitary region. This model provided valuable insights into the role of viral oncoproteins in tumor initiation and progression, particularly their interactions with key tumor suppressor proteins such as the pRB family, p53, and phosphatase PP2A [60]. The retinoblastoma T-Ag transgenic model has been invaluable for studying the origins and tumorigenesis of retinoblastoma, as well as for evaluating various preclinical therapeutic interventions. The T-Ag model does not display the focal and clonal tumors that are characteristic of human retinoblastoma. These significant drawbacks underscored the necessity of developing a conditional knockout (KO) model that accurately mimics both the origin and progression of human retinoblastoma [61].

Despite complete pRB inactivation, mice with disrupted RB1 genes do not spontaneously develop retinoblastoma like humans [62,63]. Studies have shown that compensatory mechanisms involving p107 and p130 play a crucial role in tumorigenesis [64]. GEMMs with concurrent inactivation of pRB and either p107 or p130 exhibit accelerated tumor development, highlighting the functional synergy between these family members. Notably, tumor formation is significantly enhanced in triple-knockout mice, where both pRB and p107 are deactivated along with the loss of either p130, p53, or Pten. Furthermore, in mice with the double knockout of pRB/p107 and conditionally increased MDM4 expression—a crucial negative regulator of p53, known as cancer-promoting factor—tumors progress more rapidly and aggressively [49]. The MYCN gene undergoes oncogenic mutations, such as amplification and heightened expression, even in the presence of functional RB1. Contrary to human cases where MYCN amplification drives retinoblastoma without RB1 mutation, in a mouse model, overexpressing MYCN in retinal cells with wildtype pRB did not induce tumor formation. Intriguingly, retinoblastoma only develops in mice when overexpressed MYCN collaborates with pRB loss in retinal cells. This suggests that MYCN overexpression can bypass the necessity to deactivate p107 or p130 for tumor development [65].

Cre-transgenic technology has facilitated GEMM development, enabling precise gene manipulation creating breedable pRB knockout models with the help of nestin, Chx10, and Pax-6 promoters in retinal progenitor and other cells. However, models utilizing Pax-Cre and Nestin-Cre showed limitations such as late onset, low penetrance, and non-autonomous effects on cells, hindering further investigations [66]. pRB deletion models with various Cre-transgenic lines display similar phenotypes, including mitotic figures, high cell death levels, ectopic proliferation, and photoreceptor degeneration [21]. Notably, the addition of a p107 mutation enhances several developmental phenotypes associated with retinal pRB loss, indicating functional synergy between these family members [63,67]. Moreover, the role of p130 in retinal development and tumor suppression was elucidated in different knockout models, demonstrating heterogeneous marker expression resembling retinoblastoma [64]. These models offer valuable insights into tumor progression and metastasis, facilitating the study of advanced retinoblastoma. Additionally, studies demonstrate similarities between mouse models and human retinoblastoma, particularly in molecular profiles and cell type signatures, underscoring their relevance in preclinical research [68]. The LH-beta T-Ag and p53 triple-knockout (TKO) models stand out as the top choices among transgenic Rb models for assessing both established and innovative therapeutic strategies in preclinical studies. For example, an encouraging preclinical investigation showcased the efficacy of systemic topotecan and subconjunctival carboplatin therapy in knockout mice, leading to tumor eradication and even vision restoration in some long-term survivors [61,69]. Despite their utility, GEMMs do not perfectly mimic human retinoblastoma genesis, particularly concerning the tumor cell-of-origin, which may lead to irrelevant results in therapeutic testing. While GEMMs and xenograft models each have their limitations and strengths, combining both approaches also with retinoblastoma organoids can provide a comprehensive understanding of retinoblastoma and enhance the development and testing of new therapies [45].

In recent years, *Drosophila* has emerged as a valuable model organism for cancer research due to limitations in cancer cell lines and expensive mammalian models. The *Drosophila* ortholog rbf1 shares functional similarities with RB1, particularly in regulating the G1/S transition through E2F-dependent transcription repression and Rbf1 and has been found to be pro-apoptotic in proliferative tissue [70,71]. While rbf1 mutations have minimal effects on eye development in *Drosophila*, they collaborate with the hippo tumor suppressor pathway to control cell proliferation and specification. Similarly, the Hippo pathway and RB1 in human cells are crucial for inhibiting cell proliferation by repressing E2F target genes. Clonal genetic screens identified tuberous sclerosis complex 2 (Tsc2) tumor suppressor Tsc2 and a peptidyl prolyl isomerase as critical for eliminating rbf1-deficient cells. Inactivating human Tsc2 in cancer cells inhibits RB1 mutant cell growth, indicating potential therapeutic strategies for RB1-inactivated cancers. This highlights *Drosophila*’s utility as a potential retinoblastoma model [72]. The Drosophila model seems to be valuable for initial drug screening, verifying candidates in a whole-organism context before costly rodent experiments and clinical trials. However, due to physiological differences, mammalian models like mice are still essential for establishing pharmacokinetics and pharmacodynamics. While *Drosophila* is better than in vitro cell culture for preliminary screening, it cannot fully replace mouse models [73].

The summary of the most commonly used preclinical models in neuroblastoma research is presented in Table 1.

## 3. HDACs in Retinoblastoma

The function of histones is influenced by various posttranslational modifications, including acetylation, methylation, phosphorylation, and sumoylation [78]. Histone acetylation, in particular, is tightly regulated by a dynamic interplay between histone acetyltransferases (HATs) and histone deacetylases [79]. This process modulates gene expression by changing chromatin structure, toggling between more accessible (“open”) and condensed (“closed”) states [79]. HATs add acetyl groups to lysine amino-terminal ɛ-groups on the tails of histones H2A, H2B, H3, and H4, leading to chromatin expansion and enhanced transcription factor access to DNA. Conversely, HDACs remove acetyl groups, causing chromatin compaction and transcriptional repression [80]. Both enzymes play vital roles in cellular functions, yet disruptions in histone acetylation equilibrium have been linked to tumorigenesis and cancer progression in various tumor types, including retinoblastoma [78,81].

The 18 human HDACs are classified into four classes based on their sequence homology to yeast HDAC [82] (Table 2). Class I, known as Rpd3-like enzymes, comprises HDAC1, 2, 3, and 8. Class II, resembling Hda1-like enzymes, is further divided into two subclasses: IIa (HDAC4, 5, 6, 7, and 9) and IIb (HDAC6 and 10). Class III consists of sirtuins (SIRT1-7), while Class IV solely includes HDAC11 [82]. While all HDACs possess a conserved histone deacetylase domain, they differ in their localization, structure, and expression patterns. Classes I, II, and IV share structural and sequence homology and rely on a zinc ion for their catalytic function. In contrast, Class III HDACs, functioning as NAD-dependent protein deacetylases and/or ADP ribosylases, exhibit no resemblance to other classes [82].

HDACs play diverse roles in various stages of cancer. Dysregulated expression of classical HDACs (class I, II, IV) has been implicated in a spectrum of malignancies, spanning solid tumors to hematological cancers [82]. Typically, heightened levels of HDACs correlate with advanced disease and unfavorable patient prognoses [82]. However, the impact of HDACs on cancer development may not solely hinge on their expression levels, as aberrant HDAC activity is also prevalent in tumorigenesis development [83].
ijms-25-06910-t002_Table 2Table 2Histone deacetylase classification, structures, size, cellular localization, and function in retinoblastoma (based on [84]).ClassDomain StructureSizeLocalizationFunction in RetinoblastomaIHDAC1 
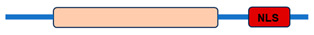
482NucleusHDAC inhibitors can differentially regulate the expression of c-Myc, a key oncogene, in retinoblastoma cell lines, and downregulate c-Myc expression in the retinoblastoma cell line WERI-Rb1 [85].HDAC2 
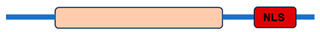
488NucleusUpregulation of HDAC2 mRNA level compared to other HDACs in WERI-Rb1 cells [85]. HDAC2 is recruited to E2F-containing promoters by interacting with the Rb protein, leading to repression of E2F-regulated genes [86]. HDAC3 
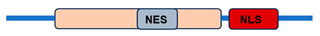

428Nucleus/CytosolN/A HDAC8 
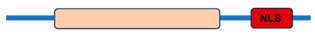

377NucleusN/AIIIIbHDAC6 
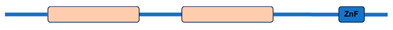
1215CytosolHDAC6 upregulated in retinoblastoma, and inhibiting HDAC6 with the selective inhibitor WT161 has anti-tumor effects on retinoblastoma cells [87]. HDAC10 
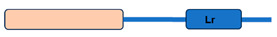

669CytosolN/AIIa HDAC4 
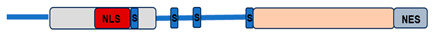

1084Nucleus/CytosolN/AHDAC5 
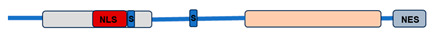
1122Nucleus/CytosolHDAC5 may play an important role in regulating the activity of the retinoblastoma protein Rb1, which is a key tumor suppressor in retinoblastoma [88]. HDAC7 
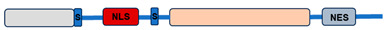

912Nucleus/CytosolN/AHDAC9 
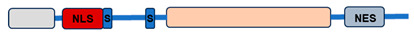
1069Nucleus/CytosolHDAC9 is upregulated in retinoblastoma. Downregulation of HDAC9 could significantly decrease cyclin E2 and CDK2 expression [89].IV HDAC11 
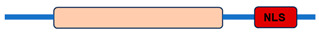

347NucleusN/AIIISIRT1
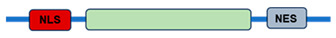
747Nucleus/CytosolUpregulated expression [90]. Silencing of SIRT1 showed that SIRT1-mediated deacetylation is essential for deactivating Rb [91]. MYCN proto-oncogene directly stimulates the transcription of SIRT1, enhancing the stability of this oncogenic protein [92].SIRT2352NucleusUpregulated expression [93].SIRT3399Nucleus/MitochondriaN/ASIRT4314MitochondriaN/ASIRT5310MitochondriaN/ASIRT6355Nucleus/CytosolUpregulated expression [93].SIRT7400NucleusN/A
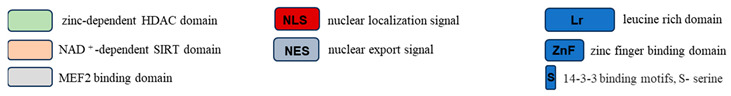
N/A—not applicable.


By inducing hyperacetylation of both histone and nonhistone substrates, HDACs can either suppress the expression of tumor suppressor genes or modulate oncogenic signaling pathways by modifying critical molecules. Individual HDACs exert regulatory effects on tumorigenesis through interactions with various cellular proteins involved in the cell cycle, e.g., cyclin-dependent kinases (CDKs), cyclins, p21, p27, retinoblastoma protein (pRb), A-kinase (or PKA)-anchoring protein (AKAP95), as well as pro- and anti-apoptotic proteins (e.g., FAS receptor/APO1-FASL ligand, tumor necrosis factor receptors (TNF-TNF), TNF-related apoptosis-inducing ligand receptors (TRAIL-TRAIL), cytoplasmic FLICE-like (cellular FLICE (FADD-like IL-1β-converting enzyme) inhibitory protein (c-FLIP), components of the DISC (death-inducing signaling complex) formation, caspase 8, Bcl-2 (B-cell lymphoma 2), Bax (Bcl-2-associated X protein), AIF (apoptosis-inducing factor), or Apaf-1 (apoptotic protease activating factor 1) [94]. HDACs and sirtuins actively participate in the DNA damage response by deacetylating histone residues such as H3K56 and H4K16, thereby regulating processes like nonhomologous end-joining (NHEJ), base excision DNA repair (BER), and double-strand break repair (DSB) [94]. They also modulate the activity of key proteins involved in these processes, including ATR (ataxia telangiectasia and Rad3-related), ATM (ataxia telangiectasia mutated), BRCA1 (breast cancer type 1 susceptibility protein), FUS (fused in sarcoma), 53BP (p53-binding protein), MSH2 (MutS protein homolog 2), PARP—poly (ADP-ribose) polymerase, and Ku70 (Lupus Ku autoantigen protein p70) [94]. Moreover, HDACs play pivotal roles in metastasis and angiogenesis by interacting with molecules such as epithelial-cadherin (CDH1), Snail, Slug (snail homolog 2), Twist (Twist-related protein 1), ZEB1 (zinc finger E-box-binding homeobox 1), ZEB2 (zinc finger E-box-binding homeobox 2), matrix metalloproteinases MPP7 and MPP8, as well as with hypoxia-inducible factors 1α (HIF-1α), heat shock proteins (HSP70 and HSP90), and vascular endothelial growth factor (VEGF) [94].

HDACs are also involved in autophagy processes in cancer [95]. Various HDAC family members exhibit both pro- and anti- autophagy activities [95], which enables autophagy-mediated cell survival by deacetylating autophagy-related 4D cysteine peptidase (ATG4D), which regulates autophagosome formation and enhances autophagy flux, thus increasing sensitivity of cells to cytotoxic drug treatment [96]. Moreover, HDACs participate in nonselective degradation by deacetylating LC3-II or, conversely, promoting the accumulation of LC3-II. Additionally, they directly deacetylate crucial regulators of the autophagy machinery, including Atg5, Atg7, and Atg8 (autophagy-related proteins) [97]. For further insights into the involvement of HDACs in tumorigenesis, refer to the review by Li and Seto [94].

Unsurprisingly, HDACs seems to be a significant player in the pathological processes of retinoblastoma. Abnormal modifications in histone acetylation are linked to the development of cancer. For instance, a common anomaly in human cancer involves the global reduction of acetylation at lysine 16 and trimethylation at lysine 20 of histone H4 [98]. Modifications to histone acetylation are facilitated by pRB protein interacting with class I HDACs (HDAC1-3) during E2F transcriptional repression, vital for cell cycle regulation and terminal differentiation [86]. The LxCxE motif within pRB’s pocket structure facilitates binding to HDACs and other chromatin remodelers [86]. HDAC1 and HDAC2 interact with the pRB/E2F complex, aiding long-term repression of E2F target genes. HDACs’ deacetylation activity is crucial for pRB-mediated repression of G1 genes, with inhibition of HDAC activity shown to hinder this process [86]. Additionally, HDACs play a role in placing repressive chromatin modifications, necessary for gene regulation. Interactions between HDAC1/2 and the pRB/E2F complex also contribute to epigenetic silencing of transposable elements, maintaining genomic stability [86]. Loss of function of the retinoblastoma gene (RB1) is a key step in initiating both hereditary and sporadic forms of retinoblastoma tumors [86]. Notably, through in vivo co-immunoprecipitation experiments and RNA interference (RNAi) silencing of SIRT1, it was demonstrated that SIRT1-mediated deacetylation is essential for deactivating Rb [91].

The amplification MYCN transcription factor, a proto-oncogene, regulates the expression of numerous genes associated with cell proliferation, survival, and metastasis [99]. Recent studies showed that c-Myc, a member of MYCN family, was downregulated in the RB cell lines WERI-Rb1 and Y79 [85]. Additionally, treatment of cells with HDAC inhibitors such as trichostatin A (TSA), vorinostat (SAHA), and entinostat (MS-275) led to a significant increase in the expression of c-Myc in WERI-Rb1 cells but not in Y79 [85]. Specifically, TSA treatment notably enhanced the activity of the c-Myc promoter in WERI-Rb1 cells. Moreover, the introduction of exogenous c-Myc resulted in a mild effect on the viability of both WERI-Rb1 and Y79 cells [85]. Furthermore, HDAC2 levels were reduced by all the other HDAC inhibitors except VPA. HDAC2 is likely implicated in the regulation of c-Myc by HDAC inhibitors. Known for its pivotal role in modulating chromatin architecture, HDAC2 leads to transcriptional changes. Moreover, upregulation of HDAC2 mRNA levels was detected compared to other HDACs in WERI-Rb1 cells. However, while HDAC2 has been reported to increase c-Myc expression in various cell types [99], this study revealed an increase in c-Myc expression when HDAC2 was downregulated, contrary to prior findings [85].

Research on Class III HDACs showed expression of SIRT1 [90], SIRT2, and SIRT6 [93] in most of the samples from retinoblastoma patients. However, the level of SIRT1 expression did not correlate with any high-risk histopathological features or affect survival outcomes. Additionally, MYCN proto-oncogene directly stimulates the transcription of SIRT1, enhancing the stability of this oncogenic protein [92]. Taken together, it may suggest a significant role of not only classical HDACs, but also the SIRT family, in retinoblastoma pathogenesis and therapy.

## 4. HDAC Inhibitors in Retinoblastoma Treatment

One of the novel approaches being investigated for treating retinoblastoma involves the use of HDAC inhibitors. These inhibitors exhibit potential in retinoblastoma treatment by altering gene expression through epigenetic mechanisms, which can specifically target cells with dysregulated E2F1 activity [13,100] (Figure 2).

HDAC inhibitors, such as trichostatin A and suberoylanilide hydroxamic acid, have demonstrated growth-inhibitory effects on retinoblastoma cells by inducing apoptosis and cell cycle arrest [13]. Additionally, these inhibitors have shown synergistic effects with standard chemotherapeutic drugs like etoposide, enhancing the sensitivity of tumor cells to treatment [13]. The specific effect of HDAC inhibitors on tumor cells, their capacity to alter gene expression patterns, and their ability to hinder the formation of new blood vessels make them highly promising for targeted therapy in retinoblastoma. This pioneering method marks a substantial stride forward in the search for more efficient and personalized treatments for this difficult childhood cancer.

Inhibitors of histone deacetylase (HDAC) have also shown promise in MYCN-amplified tumors [101]. Their impact on p27Kip1, a cyclin-dependent kinase inhibitor that plays a critical role in regulating the cell cycle, inhibits the activity of cyclin-CDK complexes, leading to cell cycle arrest [102]. This protein is tightly regulated through multiple mechanisms in various cellular processes such as cell division, differentiation, apoptosis (programmed cell death), cell movement, and metastasis. Decreased expression or mislocalization of p27Kip1 is frequently observed in many types of cancer and is associated with tumor progression and poor prognosis. p27Kip1 is considered a tumor suppressor. It can be degraded via the ubiquitin–proteasome pathway [102]. The anti-proliferative effect of HDAC inhibitors is strongly associated with increased levels of cyclin-dependent kinase inhibitor proteins (CKIs), leading to prolonged action of p27Kip1. This is due to reduced ubiquitin-dependent degradation of p27Kip1 [101,103]. It is intriguing that impairing the removal of CKIs is linked to a reduction in Skp2 dependent on HDAC inhibitors. Skp2 is a component of the ubiquitin–proteasome machinery involved in p27Kip1 degradation, which is significant in RB1-deficient retinoblastoma. This finding holds potential implications for the development of novel therapeutics, although it is constrained by the limited efficacy of Skp2 inhibitors. Increased p27Kip1 levels induced by HDACIs can lead to cell cycle arrest, differentiation, and apoptosis in various cancer cell types, including MYCN-amplified tumors [101,102].

A novel strategy also involves reducing UHRF1 activity in retinoblastoma cells to enhance their sensitivity to HDAC inhibitors. UHRF1 belongs to the RING-finger type E3 ubiquitin ligase family. It interacts with partially methylated DNA in the S-phase of cell division, bringing in the DNA methyltransferase DNMT1 to control chromatin organization and gene activity. It plays a major role in the G1/S transition of the cell cycle and regulates the expression of genes like topoisomerase IIα and the retinoblastoma gene, which are important for cell cycle progression. UHRF1 has been identified as a novel oncogene that is overexpressed in various cancers, including retinoblastoma. Its overexpression is associated with increased metastasis and poor prognosis. The protein can regulate the expression of tumor suppressor genes by mediating DNA methylation and chromatin modifications [104,105,106].

UHRF1 also plays a crucial role in regulating genes responsible for neutralizing reactive oxygen species (ROS), thereby protecting cells from apoptosis induced by HDAC inhibitors. However, UHRF1 contributes to suppressing the differentiation of photoreceptors by acting as a corepressor within a complex involving HDAC, which inhibits the expression of photoreceptor-specific genes in retinoblastoma cells. The mechanism involves UHRF1 regulating the expression of ROS-detoxifying enzymes, which protects retinoblastoma cells from the cytotoxic effects of HDAC inhibitors. When UHRF1 is depleted, the ROS-detoxifying capacity of the cells is reduced, making them more susceptible to HDAC inhibitor-induced death [107]. Furthermore, downregulation of UHRF1 enhances the antitumor effects of HDAC inhibitors in retinoblastoma cells. Depleting UHRF1 has been shown to lower the expression of GSTA4 and TXN2, genes involved in antioxidant defense, leading to an increase in basal oxidative stress. Treatment with antioxidants significantly reduces both basal and HDAC inhibitor-induced DNA damage and cell death in cells lacking UHRF1 [107]. Knocking down GSTA4 or TXN2 heightens the susceptibility of RB cells to HDAC inhibitors, highlighting their critical role in maintaining redox balance and sensitivity to HDAC inhibitor therapy following UHRF1 depletion. HDAC inhibitors exhibit varied anticancer effects, including apoptosis, cell cycle arrest, and differentiation, although the mechanisms may differ depending on the specific inhibitors and cancer cell types studied [101,103,107]. UHRF1 forms a complex with HDAC1 and binds to methylated promoters of tumor suppressor genes like CDKN2A, suggesting a collaborative role in suppressing tumor suppressors with HDAC1. Notably, HDAC inhibitors elicit stronger apoptotic responses in cells with reduced UHRF1 expression, as indicated by elevated levels of cleaved caspase-3 and poly(ADP-ribose) polymerase (PARP) [107].

The c-Myc proto-oncogene, upregulated in various cancers, plays a pivotal role in tumorigenesis. However, its expression in retinoblastoma remains unclear. c-Myc regulation involves histone acylation and DNA methylation, with different effects depending on cancer types. As a pleiotropic transcription factor, c-Myc influences cell proliferation, cell cycle progression, and programmed cell death. In retinoblastoma cell lines, c-Myc was found to be downregulated but significantly upregulated upon treatment with HDAC inhibitors. Interestingly, exogenous c-Myc reduced cell viability in retinoblastoma cells. These findings shed light on c-Myc expression and bioactivity in retinoblastoma, suggesting its potential as a therapeutic target.

One study focused on certain HDAC inhibitors (TSA, VPA, and SAHA), which were shown to significantly decrease the activity of the c-Myc promoter. In this study, treatment with HDAC inhibitors significantly upregulated c-Myc expression in WERI-Rb1, a cell line commonly used as a model for retinoblastoma research. Additionally, exogenous c-Myc reduced cell viability in both cell lines. The study suggests that histone acetylation may specifically regulate c-Myc expression in WERI-Rb1 cells. The findings underscore the potential epigenetic regulation of c-Myc in RB tumorigenesis, highlighting avenues for further research into therapeutic interventions targeting c-Myc in retinoblastoma using HDAC inhibitors [85].

HDACs play a crucial role in epigenetic regulation, influencing the expression of angiogenic factors. Angiogenesis is a critical component of tumor growth and metastasis. HDACs can modulate the chromatin architecture and transcription of genes encoding pro-angiogenic factors such as vascular endothelial growth factor (VEGF), hypoxia-inducible factor 1-alpha (HIF-1α), and CXCR4. By altering the acetylation status of histones associated with these genes, HDACs can either promote or inhibit their transcription, thereby impacting angiogenesis within the tumor microenvironment (TME). Within the hypoxic TME, tumor cells adapt to low oxygen levels by upregulating hypoxia-inducible factors (HIFs), notably HIF-1α, to promote angiogenesis. HDACs play multifaceted roles in regulating HIF-1α activity, contributing significantly to angiogenic processes [108].

One mechanism by which HDACs influence angiogenesis is through the post-translational modification of HIF-1α. Acetylation at specific lysine residues can regulate HIF-1α stability and activity. HDACs, particularly HDAC1 and HDAC4, are involved in deacetylating HIF-1α, thereby promoting its degradation via the ubiquitin–proteasome pathway. Conversely, inhibition of HDACs leads to hyperacetylation of HIF-1α, stabilizing it and enhancing its transcriptional activity, consequently promoting angiogenesis [108].

Moreover, HDACs regulate the expression and activity of molecular chaperones such as heat shock proteins (HSPs), including HSP70 and HSP90, which play essential roles in HIF-1α stabilization and function. HDAC5 and HDAC6, for instance, modulate the acetylation status of HSP70 and HSP90, impacting their interactions with HIF-1α. Hyperacetylation of these chaperones, resulting from HDAC inhibition, enhances their affinity for HIF-1α, promoting its destabilization and degradation [108]. HDACs can also directly influence the transcriptional activity of HIF-1α by regulating its association with coactivators such as CBP/p300. HDAC4, HDAC5, and HDAC7 have been implicated in promoting the recruitment of CBP/p300 to HIF-1α, thereby enhancing its transcriptional activity and consequently promoting angiogenesis [108]. In view of this, it is not unexpected that dysregulation of specific HDAC isoforms, such as HDAC9, can profoundly impact angiogenic processes. HDAC9 hyperexpression has been linked to enhanced endothelial cell proliferation and angiogenesis by suppressing the expression of antiangiogenic microRNAs within the miR-17-92 cluster. Additionally, the intricate involvement of HDACs in multiple facets of angiogenesis underscores their potential as therapeutic targets for anti-angiogenic strategies in cancer treatment. Targeting HDAC-mediated regulation of HIF-1α and associated pathways holds promise for disrupting tumor angiogenesis, thereby impeding tumor progression and metastasis [108].

WT161 is a potent and selective HDAC6 inhibitor that suppressed cell growth and induced apoptosis in retinoblastoma cells in a dose-dependent manner [87]. It showed synergistic inhibitory effects when combined with cisplatin, indicating promising efficacy for retinoblastoma treatment [87]. In breast cancer cells, WT161 demonstrated more potent growth inhibitory effects compared to the pan-HDAC inhibitors vorinostat (SAHA) and panobinostat (LBH589) [109]. This suggests that WT161 may be more effective than non-selective HDAC inhibitors. While there is no direct comparison of WT161 to other HDAC inhibitors for retinoblastoma [87], the results indicate that WT161 shows promising anti-tumor effects in retinoblastoma cells and may have advantages over non-selective pan-HDAC inhibitors based on its potency in other cancer types [109].

Promising research with pan-HDAC inhibitor belinostat was performed by Kaczmarek et al. [110]. Belinostat was equally effective as the standard-of-care melphalan in eradicating human retinoblastoma vitreous seed xenografts in a rabbit model (95.5% reduction for belinostat vs. 89.4% for melphalan), and, significantly, did not cause the severe retinal structural and functional toxicity seen with melphalan in the rabbit model [110]. At a dose of 350 μg (equivalent to 700 μg in human eyes), belinostat caused no reductions in electroretinography parameters, retinal microvascular loss, or retinal degeneration in rabbits [110]. Even at a higher dose of 700 μg (equivalent to 1400 μg in humans), belinostat caused only minimal toxicity in the rabbit model [110].

In summary, HDAC inhibitors like WT161 and belinostat have shown promising efficacy in eradicating retinoblastoma cells and vitreous seeds, with belinostat demonstrating comparable efficacy to melphalan but with a potentially better toxicity profile and reduced retinal toxicity [87,110]. While other HDAC inhibitors like vorinostat have been evaluated in some clinical trials for neuroblastoma and other cancers [111,112], there is limited direct information in these search results about the specific efficacy of vorinostat, panobinostat, or TSA for treating retinoblastoma. However, further studies are needed to fully understand the mechanisms and optimize the specificity and safety of these agents for retinoblastoma treatment. Based on the search results, histone deacetylase (HDAC) inhibitors show promising efficacy in treating retinoblastoma, but their toxicity profile needs to be carefully evaluated.

## 5. Concluding Remarks

Retinoblastoma presents significant treatment challenges due to its complexity, often requiring tailored therapeutic approaches. Dysregulated E2F1 activity, driven by genetic and molecular alterations such as mutations in the RB1 gene, plays a pivotal role in disease development by promoting uncontrolled cell proliferation and tumor formation. Targeting dysregulated E2F1 activity emerges as a crucial therapeutic strategy to inhibit cell proliferation and induce apoptosis in retinoblastoma cells.

HDAC inhibitors offer a promising avenue for retinoblastoma treatment. They demonstrate growth-inhibitory effects on retinoblastoma cells by inducing apoptosis and cell cycle arrest. Moreover, they exhibit synergistic effects with standard chemotherapeutic drugs, enhancing tumor cell sensitivity to the treatment. HDAC inhibitors hold potential for targeted therapy in retinoblastoma by altering gene expression patterns and hindering new blood vessel formation. Furthermore, HDAC inhibitors show promise in combating MYCN-amplified tumors by increasing the levels of cyclin-dependent kinase inhibitor proteins, leading to cell cycle arrest and apoptosis induction. The strategies aimed at reducing UHRF1 activity in retinoblastoma cells enhance sensitivity to HDAC inhibitors, offering another potential therapeutic approach by exploiting the interplay between HDACs and UHRF1 in retinoblastoma progression. In light of the findings discussed in this review, a thorough understanding of the intricate mechanisms involving HDACs in retinoblastoma development and angiogenesis regulation offers valuable insights into potential therapeutic strategies for improving treatment outcomes in this challenging childhood cancer.

## Figures and Tables

**Figure 2 ijms-25-06910-f002:**
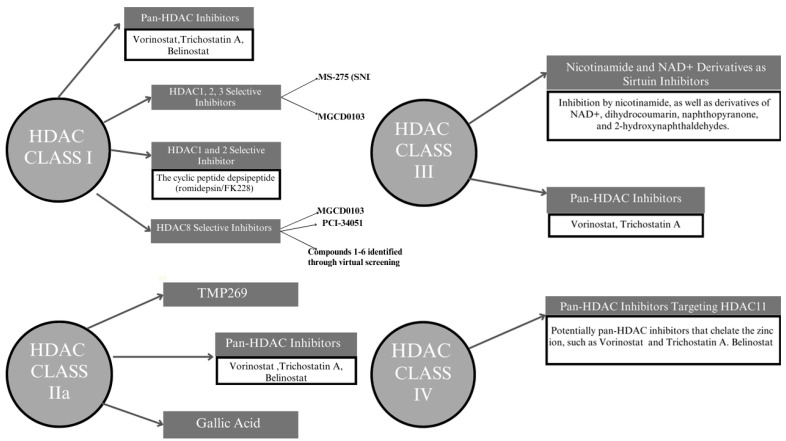
HDAC inhibitors used in retinoblastoma treatment.

**Table 1 ijms-25-06910-t001:** Characteristics of preclinical models of retinoblastoma.

Type of Retinoblastoma Model		Genetic Alterations	Laterality	Metastasis and Invasiveness	References
Cell Line	Y79	Rb1 mutation (on both alleles of RB1, including splice site mutations), MYCN amplification	uni	invasive and metastatic phenotype of RB tumor cells	[28,74]
	WERI-Rb1	loss of the entire RB1 gene with LOH (loss of heterozygosity)	uni	non-metastatic phenotype of RB tumor cells	[28,74]
	RBL13	Rb1 mutation (on both alleles of RB1, including splice site mutations)	uni	not an invasive phenotype of RB tumor cells	[29,74]
	RBL15	loss of the entire RB1 gene with LOH (loss of heterozygosity)	bi	not an invasive phenotype of RB tumor cells	[29,74]
	RBL30	Rb1 point mutations	uni	not an invasive phenotype of RB tumor cells	[29,74]
	RB247C3	Rb1 mutation (on both alleles of RB1, including splice site mutations)	bi	not an invasive phenotype of RB tumor cells	[29,74]
	RB355	Rb1 mutation (on both alleles of RB1, including splice site mutations), MYCN amplification	uni	not an invasive phenotype of RB tumor cells	[29,74]
	RB383	RB1 promoter hypermethylation, MYCN amplification	uni	not an invasive phenotype of RB tumor cells	[29,74]
	RB3823 RB522	MYCN amplification	uni	N/A	[74]
	HSJD-RBT-2 HSJD-RBVS-3	Rb1 mutation	bi	N/A	[75]
	HSJD-RBT-1 HSJD-RBVS-1 HSJD-RBT-5 HSJD-RBT-7 HSJD-RBT-8 HSJD-RBVS-8	no Rb1 mutation	uni	N/A	[75]
GEMM		*Rb1*/*p107* DKO	uni/bi	Delayed tumorigenesis of metastatic RB	[49,66,68,76]
		*Rb1*/*p130* DKO	uni/bi	Early and advanced metastatic RB	[49,63,68,76]
		*Rb1*/*p107*/*p130* TKO	uni/bi	Very aggressive metastatic RB	[49,68]
		*Rb1*/*p107*/*p53* TKO	uni/bi	Advanced and aggressive mestatic RB	[49,66,68]
		*Rb1*/*p107* DKO/*MDMX Tg*	uni/bi	Advanced and aggressive mestatic RB	[49,68]
		*Rb1*/*p107*/*Pten* TKO	uni/bi	Tumor progression related to the PI3K/AKT pathway	[49,77]
		*Rb1* KO/*MYCN*	bi	Oncogenic effects of MYCN on RB	[49,65]

uni—unilateral; bi—bilateral; N/A not available, RB—retinoblastoma.

## Data Availability

Not applicable.

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
