# Peer review of "Histone Deacetylases in Retinoblastoma"

_ijms, 2024, doi:10.3390/ijms25136910_

Round 1

Reviewer 1 Report

Comments and Suggestions for Authors

The review article "Histone deacetylases in retinoblastoma" by Lisek et al. discusses the intricate mechanisms involving HDACs in retinoblastoma development and angiogenesis regulation, and it provides valuable insights into potential therapeutic strategies for improving treatment outcomes in this tricky childhood cancer. Overall, the article is well-written and references the literature thoroughly. There is a scarcity of manuscripts that elaborate on the relationship of HDAC in retinoblastoma. There are a few comments that could help improve the review article. These are as follows:-
1. The author should add an illustration that explains the basic HDAC route and how it is related to cell-cycle proteins and retinoblastoma metabolism. This would improve the understanding of readers.
2. Figure 2 is blurry and difficult to follow.
3. The majority of the review explains the relationship between HDAC and MYCN, which accounts for fewer than 15% of retinoblastoma patients. It would be preferable if the author included a paragraph that elaborates on the interaction between HDAC and RB1 in the manuscript.

Reviewer 2 Report

Comments and Suggestions for Authors

Lisek and colleagues review the role(s) of histone deacetylases (HDACs) in retinoblastoma, including effects on cell cycle regulation and retinal ganglion cell apoptosis, and the regulation of tumor suppressors and oncogenes including RB1. They also discuss the potential of HDAC inhibitors as therapeutic agents in these ocular tumors. Much of the review covers general issues regarding retinoblastoma pathobiology and model systems rather than HDAC-specific findings, but the table on models is actually fairly useful. The review is well written overall, and fairly comprehensive, but could be improved in a number of small ways.

1) “Retinoblastoma, the most prevalent malignant intraocular tumor, predominantly affects children, originating from the cones of the retina, which possess certain traits making them susceptible to tumorigenesis.” This first sentence of the paper is incorrect in the way it is phrased – retinoblastoma is the most common malignant intraocular tumor in children. Uveal melanoma is much more common in the entire population.

2) Some terms are not defined the first time they are used. For example, ER2 in line 55 of page 2. When introducing a new acronym or gene name, the authors should make sure that a definition (or just context) is always provided.

3) The discussion of MYCN-driven retinoblastoma on page 3, line 90-93 should mention that amplification of the oncogene is the main driver of overexpression.

4) “The 18 potential human HDACs are classified into four classes based on their sequence homology to 235 yeast HDAC [68] (Table 2).” Why are these characterized as “potential” HDACs?

While the number of studies which directly address the roles of HDACs in retinoblastoma tumors is rather modest, the review does a fairly comprehensive job of bringing this work together.

Author Response

Thank you very much for taking the time to review this manuscript. Please find the detailed responses below and the corresponding corrections highlighted in grey in the re-submitted files.

Comment 1:

  • “Retinoblastoma, the most prevalent malignant intraocular tumor, predominantly affects children, originating from the cones of the retina, which possess certain traits making them susceptible to tumorigenesis.” This first sentence of the paper is incorrect in the way it is phrased – retinoblastoma is the most common malignant intraocular tumor in children. Uveal melanoma is much more common in the entire population.

Response 1: It has been corrected.

Retinoblastoma (Rb), the most prevalent malignant intraocular tumor in children, predominantly affects before the age of 3, originating from the cones of the retina, which possess certain traits making them susceptible to tumorigenesis.

Comment 2:

  • Some terms are not defined the first time they are used. For example, ER2 in line 55 of page 2. When introducing a new acronym or gene name, the authors should make sure that a definition (or just context) is always provided.

Response 2: We completely agree with Reviewer. All abbreviations have been checked and defined. Thank you especially for this comment because it was mistake instead of ER2 should be E2F.

Comment 3:

  • The discussion of MYCN-driven retinoblastoma on page 3, line 90-93 should mention that amplification of the oncogene is the main driver of overexpression.

Response 3: Thank you for your suggestion. It has been corrected:

MYCN amplification leads to overexpression, which drives tumor development and progression. MYCN-amplified retinoblastoma exhibit distinct gene signatures associated with the resultant MYCN overexpression, maintaining an undifferentiated and aggressive phenotype. Compared to the classical RB1-deficient retinoblastoma, MYCN-amplified retinoblastoma shares histopathological and genomic characteristics more similar to neuroblastoma with MYCN amplification.

 In our review, we provide additional insights from line 93 to 95, indicating that MycN-amplified retinoblastomas demonstrate unique gene signatures linked to elevated MycN expression, serving as the predominant driver in this rare and aggressive subtype.

Comment 4:

  • “The 18 potential human HDACs are classified into four classes based on their sequence homology to 235 yeast HDAC [68] (Table 2).” Why are these characterized as “potential” HDACs?

Response 4: It has been corrected.

Reviewer 3 Report

Comments and Suggestions for Authors

In this manuscript, authors have reviewed the usage of pre-clinical models and role of histone deacetylases in the progression of retinoblastoma. In addition, authors have discussed the use of HDAC inhibitors for the treatment of retinoblastoma. Overall manuscript is well written and covers all aspects. The reviewer has following suggestions to improve the manuscript.

Authors have discussed in detail about the usage of cell line models to study RB while discussion on in vivo models are limited. Utilization of both xenograft and transgenic in vivo models such as zebrafish and mice should be elaborated further. Especially, studies on employing drosophila models to characterize RB protein and mutational analysis of histone deacetylase are vast and the merits or demerits should be considered for each of these models.

Lines 304-315: It is well known that c-Myc is overexpressed in the majority of human cancers, is an oncogene known to induce cell proliferation and often denoted as diagnostic marker for cancer. On the contrary, citing previous studies authors have mentioned that c-Myc expression is downregulated in RB cell lines, treatment by HDAC inhibitors such as TSA increases c-Myc expression and exogenous c-Myc reduces cell viability. Has there been any other studies reporting increase of cMyc in RB? How different is RB progression to that of cancer?

There are other histone deacetylase inhibitors that are shown to be effective and has been studied in vivo but are not reviewed in the study such as belinostat. Both efficacy and relative toxicity as well as mode of administration such as intravitreal injections for the proposed HDACs inhibitors should be discussed.

Author Response

Thank you very much for taking the time to review this manuscript. Please find the detailed responses below and the corresponding corrections highlighted in grey in the re-submitted files.

Comment 1:

Authors have discussed in detail about the usage of cell line models to study RB while discussion on in vivo models are limited. Utilization of both xenograft and transgenic in vivo models such as zebrafish and mice should be elaborated further. Especially, studies on employing drosophila models to characterize RB protein and mutational analysis of histone deacetylase are vast and the merits or demerits should be considered for each of these models.

Response 1:

In response, we have enriched the paper by incorporating a more detailed discussion on xenograft and transgenic in vivo models, particularly in their application to preclinical testing. It is pertinent to note that both xenografts and transgenic mice faithfully replicate the histological features of human tumors and are adept at evaluating novel therapeutic strategies. However, it is imperative to recognize that each model harbors distinct advantages and limitations, rendering them complementary tools for advancing our understanding of retinoblastoma, as elucidated in this subsection of the review. Additionally, we have included information regarding other pertinent retinoblastoma models, such as the rabbit xenograft model, the rabbit orthotopic xenograft model of retinoblastoma vitreous seeds, as well as zebrafish and xenografts in the newborn rat model.

Furthermore, we have addressed the Drosophila model within the context of retinoblastoma research, underscoring the evolutionary parallels between RB1 and its Drosophila ortholog rbf1. Moreover, we have highlighted potential therapeutic avenues for cancers with RB1 inactivation, elucidating their connection to the Tsc2 tumor suppressor.

Changes have been introduced from:

Line: 129-137; 177-180; 196-197; 206-213; 222-238; 246-250; 277-301

and highlighted in grey.

Comment 2:

Lines 304-315: It is well known that c-Myc is overexpressed in the majority of human cancers, is an oncogene known to induce cell proliferation and often denoted as diagnostic marker for cancer. On the contrary, citing previous studies authors have mentioned that c-Myc expression is downregulated in RB cell lines, treatment by HDAC inhibitors such as TSA increases c-Myc expression and exogenous c-Myc reduces cell viability. Has there been any other studies reporting increase of cMyc in RB? How different is RB progression to that of cancer?

Response 2:

We haven’t find any other studies reporting c-Myc increase in retinoblastoma, however the authors of this article studied only two RB cell lines – WERI-Rb1 and Y79.  The authors tried to explain these results by:

“Moreover, the RB mutation in WERI-Rb1 cells is a complete deletion of the gene, whereas a partial deletion is present in Y79 cells. The two RB cell lines possess differing growth characteristics (43), which may contribute to the variation in their gene expression patterns and result in contrasting c-myc-silencing mechanisms. Overall, the current results demonstrated that histone acetylation of the c-myc gene was specific to WERI-Rb1 cells.

C-myc has been reported to be regulated by histone acylation or DNA methylation in numerous types of cancer cell (12–15). The present data revealed that expression levels of c-Myc were significantly upregulated in WERI-Rb1 cells following treatment with TSA, SAHA and MS-275. This result is consistent with a previous study, which revealed that TSA significantly increased the expression of c-myc mRNA resulting from nerve growth factor (NGF), and blocked both oncogenic ras- and NGF-induced neurite outgrowth from PC12 cells (23). By contrast, previous studies have indicated that c-myc may also be regulated by certain DNA demethylating reagents, such as 5-azacytidine (12–14). Certain previous studies on different cancer cells have reported that c-myc expression was downregulated or unaffected by treatment with HDAC inhibitors. For example, Kretzner et al (24) demonstrated that both TSA and SAHA decreased c-Myc mRNA and protein expression, as well as c-Myc-regulated microRNA expression. Nebbioso et al (25) reported that treatment with SAHA and MS-275 resulted in both c-Myc acetylation at lysine residue 323, and c-Myc downregulation in acute myeloid leukaemia cell lines. Furthermore, silencing of c-Myc was not influenced by TSA in prostate cancer PC3 cells (26). Thus, the present data indicated that histone deacetylation was implicated in the silencing mechanism of c-Myc in WERI-Rb1 cells.”

Oncol Lett. 2020 Jan; 19(1): 460–468.

Published online 2019 Nov 19. doi: 10.3892/ol.2019.11111

To be more specific we have added to the text:

Line 307:  led to a significant increase in the expression of c-Myc in WERI-Rb1 cells but not in Y79 [71].

Line 309: Moreover, the introduction of exogenous c-myc resulted in a significant reduction mild effect on the viability of both WERI-Rb1 and Y79 cells [71].

Comment 3:

There are other histone deacetylase inhibitors that are shown to be effective and has been studied in vivo but are not reviewed in the study such as belinostat. Both efficacy and relative toxicity as well as mode of administration such as intravitreal injections for the proposed HDACs inhibitors should be discussed.

Based on available literature we have completed information about use of belinostat and in vivo models of retinoblastoma. We haven’t found data about others HDAC inhibitor use in vivo in retinoblastoma, most articles are just predictions. We have added below paragraph about HDAC inhibitors from line 510 to 535:

“WT161 is a potent and selective HDAC6 inhibitor that suppressed cell growth and induced apoptosis in retinoblastoma cells in a dose-dependent manner [1]. It showed synergistic inhibitory effects when combined with cisplatin, indicating promising efficacy for retinoblastoma treatment [1]. In breast cancer cells, WT161 demonstrated more potent growth inhibitory effects compared to the pan-HDAC inhibitors vorinostat (SAHA) and panobinostat (LBH589) [2]. This suggests WT161 may be more effective than non-selective HDAC inhibitors. While there is no direct comparison of WT161 to other HDAC inhibitors for retinoblastoma [1], the results indicate that WT161 shows promising anti-tumor effects in retinoblastoma cells and may have advantages over non-selective pan-HDAC inhibitors based on its potency in other cancer types [2].

Promising research with pan-HDAC inhibitor belinostat was performed by Kaczmarek et al. [3]. Belinostat was equally effective as the standard-of-care melphalan in eradicating human retinoblastoma vitreous seed xenografts in a rabbit model (95.5% reduction for belinostat vs. 89.4% for melphalan), and what important did not cause the severe retinal structural and functional toxicity seen with melphalan in the rabbit model [3]. At a dose of 350 μg (equivalent to 700 μg in human eyes), belinostat caused no reductions in electroretinography parameters, retinal microvascular loss, or retinal degeneration in rabbits [3]. Even at a higher dose of 700 μg (equivalent to 1400 μg in humans), belinostat caused only minimal toxicity in the rabbit model [3].

In summary, HDAC inhibitors like WT161 and belinostat have shown promising efficacy in eradicating retinoblastoma cells and vitreous seeds, with belinostat demonstrating comparable efficacy to melphalan but with a potentially better toxicity profile and reduced retinal toxicity [1,3]. While other HDAC inhibitors like, vorinostat has been evaluated in some clinical trials for neuroblastoma and other cancers [4, 5] there is limited direct information in these search results about the specific efficacy or use of vorinostat, panobinostat or TSA for treating retinoblastoma. However, further studies are needed to fully understand the mechanisms and optimize the specificity and safety of these agents for retinoblastoma treatment. Based on the search results, histone deacetylase (HDAC) inhibitors show promising efficacy in treating retinoblastoma, but their toxicity profile needs to be carefully evaluated.”

  1. Sun J, Qian X, Zhang F, Tang X, Ju C, Liu R, Zhou R, Zhang Z, Lv XB, Zhang C, Huang G. HDAC6 inhibitor WT161 induces apoptosis in retinoblastoma cells and synergistically interacts with cisplatin. Transl Cancer Res. 2019 Dec;8(8):2759-2768. doi: 10.21037/tcr.2019.10.30. PMID: 35117033; PMCID: PMC8798655.
  2. Hideshima T, Mazitschek R, Qi J, Mimura N, Tseng JC, Kung AL, Bradner JE, Anderson KC. HDAC6 inhibitor WT161 downregulates growth factor receptors in breast cancer. Oncotarget. 2017 Jul 5;8(46):80109-80123. doi: 10.18632/oncotarget.19019. Erratum in: Oncotarget. 2021 Aug 17;12(17):1736. doi: 10.18632/oncotarget.28051. PMID: 29113288; PMCID: PMC5655183.
  3. Kaczmarek JV, Bogan CM, Pierce JM, Tao YK, Chen SC, Liu Q, Liu X, Boyd KL, Calcutt MW, Bridges TM, Lindsley CW, Friedman DL, Richmond A, Daniels AB. Intravitreal HDAC Inhibitor Belinostat Effectively Eradicates Vitreous Seeds Without Retinal Toxicity In Vivo in a Rabbit Retinoblastoma Model. Invest Ophthalmol Vis Sci. 2021 Nov 1;62(14):8. doi: 10.1167/iovs.62.14.8. PMID: 34757417; PMCID: PMC8590161.
  4. Phimmachanh M, Han JZR, O'Donnell YEI, Latham SL, Croucher DR. Histone Deacetylases and Histone Deacetylase Inhibitors in Neuroblastoma. Front Cell Dev Biol. 2020 Oct 7;8:578770. doi: 10.3389/fcell.2020.578770. PMID: 33117806; PMCID: PMC7575710.
  5. Pramanik SD, Kumar Halder A, Mukherjee U, Kumar D, Dey YN, R M. Potential of histone deacetylase inhibitors in the control and regulation of prostate, breast and ovarian cancer. Front Chem. 2022 Aug 12;10:948217. doi: 10.3389/fchem.2022.948217. PMID: 36034650; PMCID: PMC9411967.

Reviewer 4 Report

Comments and Suggestions for Authors

The paper titled "Histone deacetylases in retinoblastoma" is an update shedding the light on he epigenetic mechanisms involved in the pathogenesis of retinoblastoma with significant practical impact validated by the potential to develop innovative treatment modalities for retinoblastoma. It is useful both for researchers and clinical ophthalmologists. However, in order to correspond to the content, the title should state the fact that the paper is a review and not the result of personal research.

The introduction provides all the necessary information for the understanding of the subject, the article is written in a scientifically sound style and it cites the important and significant references for this subject.

Therefore, I consider that this paper meets the criteria for being published in IJMS.

Author Response

Thank you for your revision and suggestion about title. We also considered the title " The insight into HDAC function in retina pathobiology, but we rather prefer to keep it short. Therefore, the current title is more informative and can bring the attention of wider audience.